# Individual-differences affecting emotion regulation behaviors of injured athletes: A retrospective quantitative study

**Tomonori Tatsumi** *

Department of Education, Faculty of Education, Kio University, Umami-naka, Koryo-cho, Kitakatsuragi-gun, Nara, Japan

* t.tatsumi@kio.ac.jp

## Abstract

Suppressing the expression of negative emotions caused by injuries prevents athletes from fully accepting their sports injuries. However, acceptance is maintained and promoted when positive reappraisal and suppression are synchronized. This study analyzes individual-differences affecting the use of these two emotion regulation behaviors. A questionnaire consisting of personal variables regarding athletic identity, commitment to sports ethics and emotion regulation behavior scale for suppression and positive reappraisal was conducted on athletes with previous injuries ($N = 180$). A model estimate using correlation analysis and multiple regression analysis was examined utilizing structural equation modeling. The results indicated that commitment to sports ethics and athletic identity promoted the suppression of expressiveness toward negative emotions and positive reappraisal. In contrast, difficulty describing feelings facilitated suppression and hindered positive reappraisal. These results seem to suggest that commitment to sports ethics and athletic identity might facilitate suppression and positive reappraisal, as well as function as variables maintaining and promoting accepting sports injuries. On the other hand, difficulties in describing feelings might lead to abnormal responses caused by accumulated negative emotions due to the suppression of expressing negative emotions and lack of positive reappraisal promotion.

## Introduction

Expressing negative emotions related to injures contributes to promoting the acceptance of reality in athletes, and enables future-oriented coping behaviors, among which includes the process of rehabilitation [1–6]. Allowing injured athletes to openly express negative emotions caused by their injuries and providing an environment, i.e., space and time, which foments the expression of these emotions is essential for psychological support of injured athletes [5, 7]. However, there are issues in the world of competitive sports that must be considered in parallel with discussions about the nature of such a supportive environment. The certain climates and group norms established in athletic sports encourage athletes to perform regardless of injuries and pain [8–10]. As a result, injured athletes usually prefer not to express negative emotions

**Data Availability Statement:** All relevant data are within the paper and its Supporting Information files.

**Funding:** This work has been entirely funded by Grant-in-Aid for Scientific Research (C) 20K11380

from the Japan Society for the Promotion of
Science. The funders had no role in study design,
data collection and analysis, decision to publish, or
preparation of the manuscript. There was no
additional external funding received for this study.

**Competing interests:** The author has declared that
no competing interests exist.

that may lead to support from third parties. They tend to suppress emotions related to injuries
in order to maintain group norms and avoid negative evaluations from inside and outside the
sporting world [11]. Injured athletes might compromise their emotions and the environment
and either express or suppress emotions based on the situation. Such environment may pre-
vent people around athletes from noticing their grief, leading to insufficient support and the
accumulation of negative emotions, which in the end may cause prolonged grief, its fixation
and repetition.

On the other hand, suppression of negative emotions is not only maladaptive but also has
adaptive aspects. For example, Yoshizu [12] indicated that although emotional suppression
directly affects self-esteem and subjective well-being, these feelings might be positively main-
tained, mediated by self-control skills. Moreover, Kashimura and Iwamitsu [13] examined psy-
chological responses in patients diagnosed as having breast cancer. They indicated that the
adaptive aspects of emotional suppression in participants could temporarily distance them-
selves from their problems, prevent aggravation of problems, and protect themselves through
emotional suppression. However, emotional suppression has consistent maladaptive aspects.
Furthermore, Kimura [14] suggested that it is effective to weaken the intention to suppress
thought by referring to Wegner's Ironic Process Theory [15], which explains the paradoxical
effect of thought suppression. Kimura mentioned of a positive replacement strategy might be
effective.

Previous studies mentioned above suggest that suppression might have metacognitive func-
tions, such as developing adaptive responses by distancing themselves from the problem and
reevaluating the situation. Tatsumi and Takenouchi [6, 16] indicated negative and positive
effects of expressive suppression regarding acceptance in injured athletes; acceptance may be
maintained or promoted even when important social support is not provided if "positive reap-
praisal" with metacognitive functions synchronizes with suppression. The above findings sug-
gest the need to judge whether injured athletes' grief responses is normal or not, not only from
the perspective of suppression but also from the synchronization of positive reappraisal. Indi-
vidual variables are considered influential in the suppression of expressing negative emotions.
Therefore, examining individual variables and variables related to positive reappraisal could
help screen injured athletes that might show abnormal responses caused by suppression. Based
on the findings of preceding studies, the present analysis focused on athletes' "commitment to
sports ethics," "athletic identity," and "difficulty describing feelings" as individual variables,
and the hypothesis of their relationship with expressive suppression.

"Sports ethics" are normative content related to pains and injuries commonly experienced
in competitive sports, which many athletes adopt as the defining criteria of "real athletes" [17].
These ethics encourage athletes to perform heroic and courageous behaviors by prioritizing
social control and organizational rationality (e.g., self-sacrificing for the game or taking risks),
whereas preventing athletes from expressing negative emotions, worries, or complaints about
injuries [11]. Therefore, commitment to sports ethics has a positive correlation with suppres-
sion. On the other hand, sports ethics may also be effective in accepting injuries and making a
comeback. For example, injured athletes who deeply internalize sports ethics may perceive
injuries as something to overcome in the development of athletes positively. Here, sports ethics
function as motivation for controlling the injured self. Therefore, the degree of commitment
to sports ethics might be positively correlated with suppression and positive reappraisal. Many
athletes do not perceive excessive conformity with sports ethics as deviant and see it rather as a
criterion for confirming or reconfirm their identity as athletes [17]. Moreover, Weinberg,
Vemau, and Horn [18] indicated that athletes with a strong athletic identity tended to show
positive attitudes toward sports ethics and norms. Previous studies have indicated that athletes
with stronger athletic identity tended to have more difficulty in emotional adaptation after

injuries [19], suggesting the strong effect of negative aspects of suppression on these athletes. It is assumed that "athletic identity" has a positive correlation with expressive suppression. On the other hand, injured athletes with stronger athletic identities have stronger conformity with sports ethics, suggesting a positive correlation with positive reappraisal enabling them to control themselves in injured situations.

"Difficulty describing feelings" refers to a personal tendency making it difficult to describe or convey self-feelings to others, which has been mentioned as an aspect of alexithymia [20]. Moreover, Tatsumi and Fukumoto [21] indicated that difficulty describing feelings was positively correlated with avoidance coping and negatively correlated with positive-thinking coping. Difficulty describing feelings might hinder clear expressions of negative emotions and emotional differentiation through expression. The above characteristics of difficulty in describing feelings assume a positive correlation with suppression and a negative correlation with positive reappraisal.

Based on the above considerations, the following hypotheses were developed in this study.

**Hypothesis 1.** Commitment to sports ethics and athletic identity enables athletes to use suppression and positive reappraisal. Concretely, commitment to sports ethics and athletic identity has positive relationships with suppression and positive reappraisal.

**Hypothesis 2.** Difficulty describing feelings enables athletes to use suppression instead of positive reappraisal. Concretely, difficulty expressing feelings has a positive relationship with suppression, whereas having a negative relationship with positive reappraisal.

To test the above hypothesis, the present study quantitatively examined the relationship between individual variables related to commitment to sports ethics, athletic identity, difficulty describing feelings, and variables of emotion regulation behavior, including expressive suppression and positive reappraisal.

## Methods

### Study design

Injured athletes' emotion regulation behaviors change with changes in their rehabilitation process. Nevertheless, it might be possible to identify the influence of individual variables on emotion regulation behaviors by carefully reviewing the past rehabilitation process of athletes with injury experiences. Therefore, this study was designed as a questionnaire survey-based quantitative study using the retrospective method, and examined individual variables that predict emotion regulation behaviors based on two hypotheses.

### Participants and procedures

This study focused on past injury experiences of competitive athletes. When using questionnaire surveys to examine past injury experiences, injury experiences in the relatively recent past should be targeted to reduce memory problems caused by recall biases. In addition, the athletes' movements athletes immediately after the rehabilitation period and the athletes' current injury status might distort their responses to questionnaires. Therefore, athletes who are currently injured or recently returned to sports should be excluded from investigations. Based on these considerations, this study's participants met the following eligibility criteria proposed by Tatsumi [22]. (a) Former-injured athletes who were forced to stop sports for over a week for rehabilitation due to an injury. (b) Athletes that return to sports in under one week and not currently undertaking rehabilitation. (c) Student-athletes in athletic clubs with good competitive results who intend to participate in competitive sports. (d) Athletes who have experienced

injuries other than sports, and injuries or diseases not treated by orthopedic surgeons, such as head injuries or physical illnesses, were excluded from this study.

The purpose of the study was explained to researchers specializing in psychology or athletic rehabilitation, enrolled at two universities with sports science faculties and one university conducting classes consisting of sports referrals, and advisors of five strong athletic clubs of three universities. Consent was obtained for cooperation in the survey, which included the distributing and collecting questionnaires. The questionnaire included documents explaining the purpose and procedures of the survey along with a document consisting of a request for cooperation and a consent form in an envelope for the participants. Participants responded to the questionnaire anonymously, sealed them, and submitted them by themselves. The above procedures were taken to preserve the anonymity of the participants. Participants that did not agree to participate in the survey submitted blank documents following the same procedure.

The questionnaire was distributed to 205 participants, all of which were collected. The number of valid respondents, which excluded a total of 25 respondents (13 men and 12 women) who did not meet the eligibility criteria and who have missing values or a biased in response tendency, was 180 (70 men and 110 women, mean age = 20.27, $SD$ = 1.02). Table 1 shows the basic information of the participants. 95 participants (52.78%) are engaged in individual sports, most of which included soccer 32 (17.78%), basketball 31 (17.22%), handball 16 (8.89%), and softball 10 (5.56%). In addition, 85 participants (47.22%) are engaged in group sports, and track and field 19 (10.56%), judo 19 (10.56%), soft tennis 12 (6.67%), kendo 12 (6.67%), and wrestling 9 (5.00%). The athletic performance levels of the participants included 55 (30.56%) participants in international tournaments~higher than Rank 4 of national tournament teams, 56 (31.11%) in rank 5–8 in national tournaments teams, 68 (37.78%) in rank 9 teams, or participating in national tournaments, and one (.56%) in a team not participating in national tournaments but in a higher-ranking regional league. Therefore, the performance

**Table 1. Basic information on the participants.**

| Characteristics | $n$ (%) |
| --- | --- |
| Age (years) | |
| $M$/$SD$/Median/Range | 20.27/1.02/20/18-24 |
| Sex | |
| Male | 70 (38.89) |
| Female | 110 (61.11) |
| Competition Types | |
| Individual (mostly including Track and Field, Judo, Soft tennis, Kendo, Wrestling) | 95 (52.78) |
| Groupe (mostly including Football, Basketball, Handball, Soft ball) | 85 (47.22) |
| Competition Levels of Team | |
| Not participating in national tournaments but in a higher-ranking regional league | 1 (.56) |
| Rank 9 or participants in national tournaments | 68 (37.78) |
| Rank 5–8 in national tournaments | 56 (31.11) |
| International tournament~higer than Rank 4 of national tournaments | 55 (30.56) |
| Degree of Injury (number of days of restricted activity) | |
| $M$/$SD$/Median/Range | 66.58/87.13/31/7-547 |
| Injury Types | |
| Rupture or injury to muscles or ligaments | 155 (86.11) |
| Fracture | 20 (11.11) |
| Joint dislocation | 5 (2.78) |

**Notes:** Injury types may overlap from a medical point of view, but are based on the description.

levels of the participants were relatively high. The types of sports injuries they experienced included rupture or injury to muscles or ligaments: 155 (86.11%), fracture: 20 (11.11%), and joint dislocation: 5 (2.78%). Ten participants did not report the number of days during which they were instructed to stop sports because of injuries. However, based on the diagnosis the injury required at least one-week suspension from the injury's date to the comeback. The average number of days of sports suspension excluding these 10 participants was 66.58 ($SD$ = 87.13), the median was 31, which indicated a left-inclined distribution.

The survey was conducted from November to December in 2014. Approval for the study was obtained from the author's research institute's ethics committee before commencing the study (approval number H26-23).

## Measures

**Commitment to sports ethics.** Hughes and Coakley [17] mentioned four factors composing sports ethics; self-sacrifice for the game, striving for distinction, risk-taking, and refusing to accept limits. A provisional scale consisting of four items based on descriptions that "athletes are required to sacrifice and be calm and cool" related to "risk-taking" known to be correlated with the attitudes toward injuries and pains was developed. These items consisted of, "it is important to continue playing for the team even injured," "players should be tough to keep performing by withstanding pains," "players should never show the pain of injury to others," and "players should never show their weakness to others."

**Athletic identity.** The Japanese version of athletic identity assessment scale developed by Isogai, Brewer, Cornelius, Etnier, and Tokunaga [23], consisting of seven items was used. One item regarding injury, "I would be very depressed if I cannot do sports because of injury," was not adopted because the present study directly examined injury situations, which might distort the descriptions. Cronbach's coefficient alpha was .80, which confirmed the reliability of the 6-item scale.

**Difficulty describing feelings.** A subscale of the Toronto Alexithymia Scale-20 (TAS-20) developed by Bagby, Parker, and Taylor [24] as well as Bagby, Taylor, and Parker [25] was used. Cronbach's coefficient alpha was .85, which confirmed the reliability of the scale.

**Emotion regulation behaviors.** Subscales related to expressive suppression (four items) and positive reappraisal (four items) of the Emotion Regulation Behavior Scale developed by Tatsumi and Takenouchi [6] based on Nozaki [26] as well as Kashimura and Iwamitsu [13] were used. Cronbach's coefficient alpha was .82 and .81, which confirmed the reliability of the scale.

Participants were asked to recall when they got injured and respond to the Athletic Identity Scale and four items of expressive suppression of the Emotion Regulation Behavior Scale using a seven-point Likert scale (1: strongly disagree~7: strongly agree). They also responded to the Sports Ethics Commitment Scale (four items), the Difficulty Describing Feeling Scale (five items), and four items of the positive reappraisal of the Emotion Regulation Behavior Scale using a five-point Likert scale (1. Strongly disagree~5. Strongly agree).

## Statistical procedures

A factor analysis of the provisional item scores from the sports ethics commitment Scale was conducted to examine the scale's factorial validity. The reliability of the scale was examined using Cronbach's coefficient alpha. The criterion for reliability corresponds with Cronbach's coefficient of alpha .70 or higher. Next, correlations between three individual variables and two subscales of the emotion regulation behavior scale were examined through correlation and multiple regression analysis. The sports ethics commitment scale, athletic identity scale,

difficulty describing feelings scale, and subscale scores of the Emotion Regulation Behavior Scale were calculated by dividing each scale's total score by the number of items. Multiple regression analyses were conducted using the forced entry method and estimated an initial model based on the standardized partial regression coefficient and the regression formula's significance. Finally, the fit of the initial model was estimated using structural equation modeling. Indices of the goodness of fit of the model were CMIN, Goodness of Fit Index (GFI), Adjusted Goodness of Fit Index (AGFI), Comparative Fit Index (CFI), Root Mean Square Error of Approximation (RMSEA), and Akaike's Information Criterion (AIC). Finally, the model with the highest goodness of fit was adopted. The IBM SPSS 20.0 Statistics and Amos 20.0 was used for the statistical analyses with the alpha level of .05. for rejecting the null hypothesis.

## Results

### Development of a scale

Exploratory factor analysis (principal factor method, Promax rotation) was conducted on four Sports Ethics Commitment Scale items. The factors were extracted based on the Kaiser-Guttman criterion, which includes eigenvalues greater than 1.0 and the absolute value of factor loadings of more than .40. The eigenvalue decay was 2.40, .85, and .40. A one-factor solution was adopted which consisted of a factor loadings ranged from .62 to .73, and Cronbach's coefficient alpha was .78, indicating the scale's reliability. On the other hand, values of goodness-of-fit indices obtained through confirmatory factor analysis were below; $\chi^2(2) = 39.798$, $p = .000$, GFI = .898, AGFI = .492, CFI = .828, RMSEA = .325, AIC = 55.798, suggesting insufficient goodness of fit of the one-factor model. The initial eigenvalue criterion and conducted exploratory factor analysis was revised with two factors. Consequently, two factors corresponding to the assumed theory were extracted when developing the scale: (1) "Calm and cool" and (2) "Self-sacrifice". The inter-factor correlation value was .60, indicating a relatively strong correlation. Factor loading was more than .80 for the first factor and more than .76 for the second factor. Cronbach's coefficient alpha was .79 for the first factor and .75 for the second factor, indicating the subscales' reliability. Finally, confirmatory factor analysis was conducted using the two factors. The results are shown in Fig 1. The goodness-of-fit indices were significantly good; $\chi^2(1) = .092$, $p = .761$, GFI = 1.000, AGFI = .997, CFI = 1.000, RMSEA = .000, AIC = 18.092. It was determined that the two-factor model was valid for the current study. As described above, the inter-factor correlation was relatively strong. As a result, the composite scores of two factors in subsequent analyses were used.

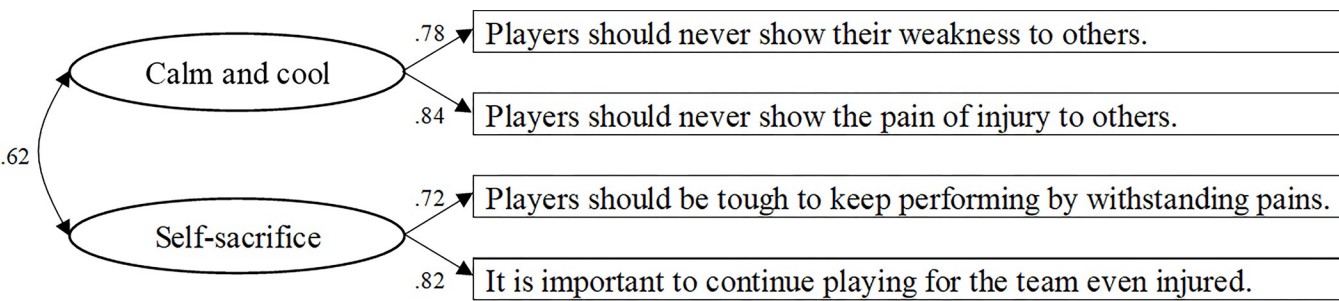

**Fig 1. Results of confirmatory factor analysis of sports ethics commitment scale** (*N* = 180).

## Correlations between individual variables and emotion regulation behaviors

Firstly, the correlations among three scales related to individual variables were examined; athletic identity, sports ethics commitment, difficulty describing feelings, and two scales of emotion regulation behaviors, expressive suppression, and positive reappraisal. The upper part of Table 2 shows basic statistics of each variable and the results of correlation analysis. Significant correlations were indicated between three individual variable and expressive suppression scales (athletic identity: $r = .29$ $p = .000$; sports ethics commitment: $r = .37$ $p = .000$; difficulty describing feelings: $r = .29$; $p = .000$). Significant correlations were indicated between athletic identity as well as sports ethics commitment and positive reappraisal ($r = .22$, $p = .003$; $r = .23$, $p = .002$). However, difficulty describing feelings and positive reappraisal were not correlated ($r = -.07$, $p = .325$).

In addition, the correlation coefficient between these variables was calculated to confirm whether age, sex, competition types, competition levels, and degree of injury (number of days of restricted activity prescribed by a doctor) were related to each scale. The results are shown in the lower part of Table 2. Sex was converted into a dummy variable with male as 0 and female as 1. Similarly, competition types were converted into a dummy variable with individual competition as 0 and group competition as 1. Regarding the dummy variables for the competition levels, the participants belonging to the lowest-level team shown in Table 1 were converted to 0, and the participants belonging to the highest-level team were converted to 3. As a result, only the competition level and the positive reappraisal showed a significant correlation in relation to the test of the hypothesis, and no correlation was shown between the other variables. Based on this result, a partial correlation coefficient was calculated using the competition level as a control variable, but there was almost no change from the correlation coefficient shown in Table 2 (Subtracting the partial correlation coefficient from the correlation coefficient, resulted in the range of -.009-.014). Therefore, the subsequent analysis was conducted without controlling age, sex, competition types, competition levels, and degree of injury.

**Table 2. Descriptive statistics and intercorrelations of the major study variables ($N = 180$).**

| | M | SD | 95% CI Lower | 95% CI Upper | 1 | 2 | 3 | 4 | 5 | 6 | 7 | 8 | 9 |
|---|---|---|---|---|---|---|---|---|---|---|---|---|---|
| 1. Athletic Identity | 6.00 | .83 | 5.87 | 6.12 | — | | | | | | | | |
| 2. Sports Ethics Commitment | 3.39 | .86 | 3.26 | 3.51 | .23 ** | — | | | | | | | |
| 3. Difficulty Describing Feelings | 3.08 | .92 | 2.95 | 3.22 | .04 | .18 * | — | | | | | | |
| 4. Suppression | 4.46 | 1.26 | 4.27 | 4.64 | .29 *** | .37 *** | .29 *** | — | | | | | |
| 5. Positive Reappraisal | 3.64 | .88 | 3.52 | 3.77 | .22 ** | .23 ** | -.07 | .24 ** | — | | | | |
| 6. Age | 20.27 | 1.01 | 20.12 | 20.42 | .00 | -.06 | -.06 | .07 | .05 | — | | | |
| 7. Sex (Ref: Male) | | | — | | .13 | .38 *** | .10 | .10 | .09 | -.10 | — | | |
| 8. Competition Types (Ref: Individual) | | | — | | .16 * | .06 | .07 | .08 | -.08 | -.05 | .26 ** | — | |
| 9. Competition Levels (Ref: Low) | | | — | | -.04 | -.07 | -.06 | -.07 | .15 * | .07 | .21 ** | -.27 *** | — |
| 10. Degree of Injury | 66.58 | 87.13 | 53.48 | 79.68 | .01 | -.08 | .10 | .00 | .01 | .07 | .11 | .15 * | -.07 |

**Notes:** CI = Confidence Interval.

*$p < .05$

**$p < .01$

***$p < .001$.

**Table 3. Results of multiple regression analysis.**

|  | Suppression | | | Positive Reappraisal | | |
| --- | --- | --- | --- | --- | --- | --- |
|  | β | p | 95% CI | β | p | 95% CI |
| Athletic Identity | .21 | .002 | [.12, .52] | .18 | .015 | [.04, .34] |
| Sports Ethics Commitment | .28 | .000 | [.22, .61] | .21 | .005 | [.07, .37] |
| Difficulty Describing Feelings | .24 | .001 | [.14, .50] | -.12 | .103 | [-.25, .02] |
| $R^2$ | .24 | | | .10 | | |
| F | 18.05 | .000 | | 6.43 | .000 | |

Next, the model paths between individual variables and emotion regulation behaviors were estimated using multiple regression analysis (see Table 3). Significant positive standardized partial regression coefficients were identified from sports ethics commitment, difficulty describing feelings, and athletic identity to expressive suppression ($\beta$ = .28, $p$ = .000; $\beta$ = .24, $p$ = .001; $\beta$ = .21, $p$ = .002, respectively). Moreover, significant positive standardized partial regression coefficients were shown from sports ethics commitment and athletic identity to positive reappraisal ($\beta$ = .21, $p$ = .005; $\beta$ = .18, $p$ = .015). However, the standardized partial regression coefficient from difficulty describing feelings to positive reappraisal was not significant ($\beta$ = -.12, $p$ = .103).

Finally, the initial model with three individual variables as exogenous variables and expressive suppression, as well as positive reappraisal as endogenous variables was examined using structural equation modeling. A covariance among exogenous variables was taken into consideration because significant correlations between sports ethics commitment as well as athletic identity and difficulty describing feelings were identified ($r$ = .23, $p$ = .002; $r$ = .18, $p$ = .017). In addition, a covariance between expressive suppression and positive reappraisal errors was taken into consideration. Consequently, the values of goodness-of-fit indices of the initial model were good; $\chi^2(2)$ = 3.015 ($p$ = .222), GFI = .993, AGFI = .950, CFI = .987, RMSEA = .053, AIC = 29.015. A path from difficulty describing feelings to positive reappraisal was established, which was not included in the initial model. The results of the analysis indicated better goodness-of-fit indices than the initial model; $\chi^2(1)$ = .309 ($p$ = .578), GFI = .999, AGFI = .990, CFI = 1.000, RMSEA = .000, AIC = 28.309, suggesting that the second model explained the data better than the initial model. Fig 2 shows the results of the analysis. All the path coefficients in

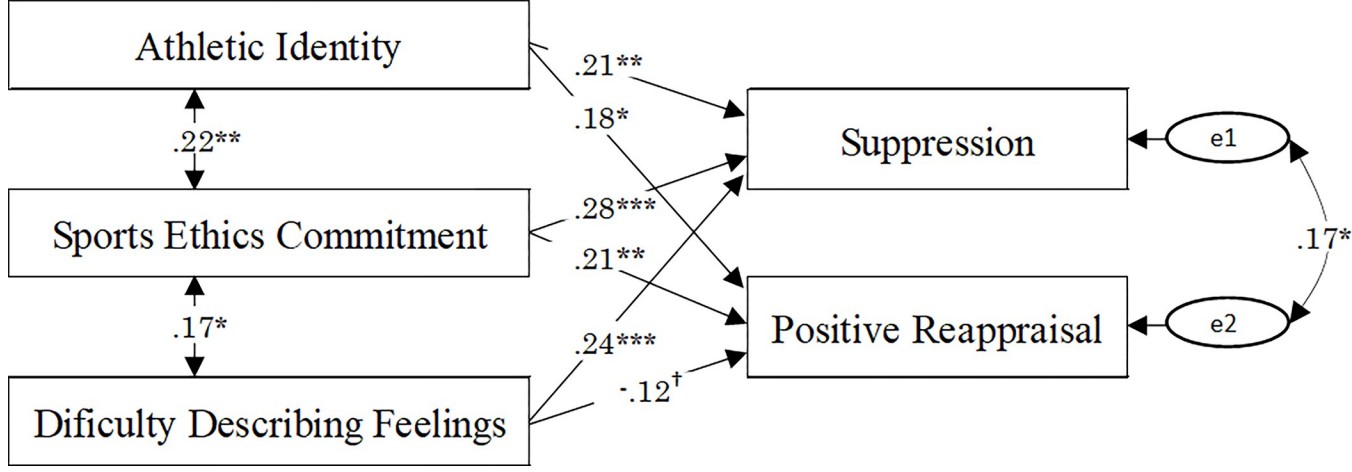

**Fig 2. Results of structural equation modeling.** †$p$ < .10, *$p$ < .05, **$p$ < .01, ***$p$ < .001.

Fig 2 were significant ($p$ = .000−.013) except for the path coefficient from difficulty describing feelings to positive reappraisal ($p$ = .098), which was only marginally significant.

The individual variables of sports ethics commitment, difficulty describing feelings, and athletic identity described positive paths to expressive suppression (path coefficients; .28; 24; 21). Sports ethics commitment and athletic identity described positive paths to positive reappraisal (.21; .18), whereas difficulty describing feelings described a negative path to positive reappraisal (-.12). Moreover, there were positive correlations between sports ethics commitment and athletic identity and difficulty describing feelings (.22;.17).

In summary, firstly, it seems possible to point out that all three individual variables of sports ethics commitment, difficulty describing feelings and athletic identity have a positive effect of facilitating the use of expressive suppression for negative emotions. Secondly, it seems possible to conclude that sports ethics commitment and athletic identity have a positive effect of facilitating the use of positive reappraisal as well. On the other hand, and thirdly, it seems possible to point out that difficulty describing feelings has a negative effect on preventing the use of positive reappraisal. Fourth, regarding personal variables, sports ethics commitment is positively related with both athletic identity and difficulty describing feelings. As a resulted, the above examination substantiates the two hypotheses developed in the present study.

## Discussion

The results of the present analysis suggest a direct relationship with the hypotheses laid out in the introduction and can be summarized in following: firstly, the results indicate that all three individual variables might facilitate the suppression of expressing negative emotions. Secondly, sports ethics commitment and athletic identity might facilitate positive reappraisal. Thirdly, difficulty describing feelings hindered positive reappraisal. Fourthly, sports ethics commitment was positively correlated with athletic identity and difficulty describing feelings.

Suppressing negative emotions after being injured prevents athletes from recovering from their emotional stability and accepting the reality of their injury. However, synchronizing positive reappraisal with suppression can lead to maintaining or promoting the acceptance of athletic injuries [6, 16]. This study focused on suppressing negative emotions after being injured, positive reappraisal, three individual variables that potentially correlated with emotion regulation behaviors and the examinations of correlations among variables using structural equation modeling. The results indicated that sports ethics commitment and athletic identity had a positive correlation with suppression and positive reappraisal. In contrast, difficulty describing feelings had a positive correlation with suppression and a negative correlation with positive reappraisal, which supported Hypotheses 1 and 2 of this study.

Athletes that have internalized sports ethics might prioritize maintaining group norms and order more than processing personal emotions. Therefore, they might try to conform to their environment, even being injured by suppressing the expression of negative emotions. On the other hand, sports ethics require each athlete to be developmentally and physically hardy and have self-control. Therefore, athletes with a high commitment to sports ethics might make positive reappraisal even after being injured to overcome this critical situation.

It has been noted that athletes with a solid athletic identity tended to have difficulties in emotional adaptation after being injured [19], suggesting that athletes with stronger athletic identity would have stronger feelings of loss, frustration, and crisis. On the other hand, such feelings might also motivate injured athletes to recover their identity and overcome difficult situations. High athletic identity might improve or maintain positive efforts using expressive suppression and positive reappraisal during the recovery from injury. It is suggested that commitment to sports ethics and athletic identity promote the acceptance of injury.

On the other hand, difficulties in describing feelings had a negative effect through promoting suppression and preventing positive reappraisal. Injured athletes who are not good at describing or conveying their feelings to others had difficulties expressing their worries and complaints about an injury. These athletes might try to adapt themselves to the competitive environment by suppressing their negative emotions. Difficulty in describing feelings is one factor in evaluating and diagnosing alexithymia tendency, a personality trait in which one expresses more interest in external issues than one's inner world [27]. Alexithymia is related to immature defense mechanisms in stressful conditions [28–32]. Injured athletes with more difficulties in describing feelings could also face more difficulties in making positive reappraisal. Studies on coping with stress related to injuries have indicated that difficulties in describing feelings might have indirect adverse effects on accepting sports injuries, which is negatively mediated by positive thoughts [21]. Difficulties in describing feelings might accumulate negative emotions through expressive suppression and cause abnormal responses such as prolonged grief and the fixation and the repetition of grief. It helps examine psychosocial support for injured athletes having more difficulties in describing feelings from a therapeutic perspective. Moreover, educational support should be provided for these athletes' emotional and behavioral problems from a preventive perspective.

There was a significant covariance between commitment to sports ethics and athletic identity and difficulty describing feelings, which suggest positive and negative aspects of conformity to sports ethics. For example, it was positively correlated with developing an athletic identity and developing difficulties in describing feelings. Hughes and Coakley [17] defined excessive conformity to sports ethics, including making sacrifices by getting injured or continuing performance without recovering, as "positive deviance," and warned that doctors and trainers supported such sports ethics and promoted positive deviance of athletes. Promoting positive deviance in the athletes' environment might improve their commitment to sports ethics and contribute to forming an athletic identity. However, it might also produce injured athletes with difficulties in expressing their negative emotions by considering group norms.

## Conclusions

In conclusion, injured athletes with difficulties describing feelings tend to suppress negative emotional expressions and are more likely to manifest abnormal responses. Therefore, practitioners attending to injured athletes should carefully monitor their emotional response tendencies, including whether they fixate on or repeat adverse emotional reactions. It would be efficacious to plan psychological interventions focusing on expressing negative emotions for these athletes. At the very least, injured athletes who tend to suppress negative emotional expressions should not be left alone and isolated because it leads to suppressing emotional expressions. They might lose opportunities to express their negative emotions due to their isolation.

In addition, the strength of athletic identity and sports ethics make athletes' post-injury responses more negative. However, they also have positive effects, such as facilitating recovery-related thoughts and behaviors. Therefore, successful interventions should not be directed at simply controlling these factors.

Nevertheless, the competitive sports ethics' role in causing difficulties athletes face in describing their feelings need to be examined. Such investigations are expected to provide helpful information for treating injured athletes with abnormal responses and preventing adverse post-injury reactions. The information provided by such studies will also help solve general clinical problems that require crisis intervention, in addition to athletic injuries.

## Limitations of the present study

Based on the ideas of Cavan and Ferdinand [33] and Dodge [34], Hughes and Coakley [17] described positive deviance as excessive conformity to sports ethics and behaviors falling outside the mid-range of the normal distribution curve. In other words, there may be negative deviance, i.e., nonconformity to sports ethics, as well as normal deviance, which is in the middle range of the distribution. Yamamoto [35] summarized the above ideas as the critical normal distribution approach. The present study confirmed a linear correlation between commitment to sports ethics and emotion regulation behaviors and examined its relations. From the perspective of the critical normal distribution approach, over-conformity to sports ethics might affect the development of difficulties in describing athletes' feelings. Nevertheless, this study did not examine concrete correlations among individual variables.

Injured athletes in rehabilitation may face stagnation of their athletic ability. Studies on burnout in athletes have reported the dangers of clinging to coping behaviors and the obsession with targets [36, 37]. Moreover, Kishi [38] regarded foreclosure of identity directed at only sports success and identity in the single dimension as problematic. The present study emphasized the positive effects of athletic identity on emotion regulation. However, Kishi [38] suggested that injured athletes' identity should be perceived from multifaceted dimensions instead of merely the athletic identity. It would be necessary to examine how injured athletes' identity is maintained and how sports ethics affect athletic identity formation.

This study examined the relationship between individual variables and emotion regulation behaviors from a macroscopic perspective and obtained specific clinical findings. However, this study only clarified the theoretical tendency based on the hypothesis. To refine the findings, the study must be conducted with a design that sufficiently controls the influence of participant's memory bias and psychological state at the time of the survey on the recall of emotional experiences. In addition, emotion regulation predominantly used by injured athletes change over time in relation to the nature of distress and emotional responses [39]. In other words, it would be assumed that the predominantly functioning personal variables and emotion regulation behaviors change over time. As a result, future studies should follow injured athletes longitudinally and examine the time series of relationships between variables during normal and abnormal responses.

Further longitudinal qualitative examinations based on individual cases would be required to clear above issues.

## Supporting information

**S1 File.**
(XLSX)

## Acknowledgments

I wish to thank all the athletes who have participated in this research.

## Author Contributions

**Conceptualization:** Tomonori Tatsumi.

**Data curation:** Tomonori Tatsumi.

**Formal analysis:** Tomonori Tatsumi.

**Funding acquisition:** Tomonori Tatsumi.

**Investigation:** Tomonori Tatsumi.

**Methodology:** Tomonori Tatsumi.

**Project administration:** Tomonori Tatsumi.

**Validation:** Tomonori Tatsumi.

**Visualization:** Tomonori Tatsumi.

**Writing – original draft:** Tomonori Tatsumi.

**Writing – review & editing:** Tomonori Tatsumi.

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
