## [Decision Letter · Decision Letter 0]

9 Feb 2022

PONE-D-21-17660

Individual-differences affecting emotion regulation behaviors of injured athletes: A retrospective quantitative study

PLOS ONE

Dear Dr. Tatsumi,

Thank you for submitting your manuscript to PLOS ONE. After careful consideration, we feel that it has merit but does not fully meet PLOS ONE’s publication criteria as it currently stands. Therefore, we invite you to submit a revised version of the manuscript that addresses the points raised during the review process.

The manuscript has been evaluated by three reviewers, and their comments are available below.

The reviewers have raised a number of major concerns. They request additional information on methodological aspects of the study (such as the exclusion criteria) and revisions to the statistical analyses.

Could you please revise the manuscript to carefully address the concerns raised?

We look forward to receiving your revised manuscript.

Kind regards,

Lorena Verduci

Staff Editor

PLOS ONE

Journal Requirements:

2. Thank you for stating in your Funding Statement: "The work has been supported in part by Grant-in-Aid for Scientific Research (C) 20K11380 from the Japan Society for the Promotion of Science. The funders had no role in study design, data collection and analysis, decision to publish, or preparation of the manuscript. No additional external funding was received for this study."

Reviewers' comments:

Reviewer's Responses to Questions

**Comments to the Author**

1. Is the manuscript technically sound, and do the data support the conclusions?

Reviewer #1: No

Reviewer #2: Partly

Reviewer #3: Yes

2. Has the statistical analysis been performed appropriately and rigorously? 

Reviewer #1: No

Reviewer #2: Yes

Reviewer #3: Yes

3. Have the authors made all data underlying the findings in their manuscript fully available?

Reviewer #1: Yes

Reviewer #2: No

Reviewer #3: No

4. Is the manuscript presented in an intelligible fashion and written in standard English?

Reviewer #1: Yes

Reviewer #2: Yes

Reviewer #3: Yes

5. Review Comments to the Author

Reviewer #1: The research addresses a relevant topic namely the emotional responses people feel when injured. The authors make speculative hypothesis regarding the emotional impact of injury and seek to test these hypotheses among a group of student athletes. There is some merit in the research method that is followed. However, there are some limitations regarding the conceptual framework underpinning the work. The first part is the notion that the emotional response to injury is fixed and not transient. At different stages of an injury people experience different emotions. The research design does not accommodate this.

Second limitation is the suggestions that causal relationships are being tested. All data were collected at the same time and therefore the possibility that the direction of relationships goes both ways is plausible.

When these two factors are taken into account, it undermines the believability and contribution of the work considerably. It is possible to re interpret the data and treat the study that is exploring relationships. This represents a very different type of study and one that is more suitable in a different journal.

Reviewer #2: I thank the authors for the work done. Despite being a cross-sectional study, the manuscript is well developed, however, some adjustments are necessary to improve the presentation of the results, therefore I suggest the following:

- It would be interesting to add the characteristics of the sample: how many female? what sports did they practice? at what level?

- Please incorporate study design

- Add the registration number of the ethics committee.

- On line 247 please add the p-value in each variable.

- Were analyzes carried out distributing the group by sex, age, sport or competitive level? I think it would be important to clarify it.

- The regression models were adjusted for some factor (age, sex, etc.)

- Although the discussion is appropriate, it is necessary to contrast the literature according to the characteristics of the sample (sports practiced, competitive level, etc).

- I think it is possible to add a conclusion

Reviewer #3: This is an interesting article which aimed to examining individual variables and variables related to positive reappraisal what should help screen injured athletes that might show abnormal responses caused by suppression.

The theme of the article is relevant because still little is known about relationship between individual variables and emotion regulation behaviors.

Whilst the article is interesting, several points are worth addressing.

Abstract:

Line 19 - add measured variables, add information what kind of relationship was evaluated and which statistical methods were used (e.g. correlations, regression).

Introduction:

General comment: use impersonal form permanently (e.g. Following hypotheses were developed ... instead of I developed the following hypotheses....).

Line 74 - should be relationship instead of correlations (regression was also used in analysis).

Line 109 - delete heading

Lines 70-76 - should be combine with established hypotheses at the end of Introduction paragraph

Methods:

Line 131 - exclusion criteria should be presented

above in relation to line 144 - how many participants (men and women) didn't meet each criterion

line 143 - questionnaire should be shortly described; was the questionniaire earlier published (citation needed) or it was authors survey (information about validation procedure needed). What kind of injuries were registered? It should be explained.

Line 194 - Heading: Statistical procedures instead of Analysis method

Line 197 - Interpretation of Cronbach alpha with internal consistency range (rules of thumb) should be presented.

Results:

Line 239 - Basic descriptive statistics of the variables used in article should be calculated (mean, sd, 95% CI) and presented in first table.

Discussion:

Discussion is generally well-written. The summary, however, lacks any specific information as to whether the hypotheses have been rejected or accepted.

References:

References matched the problem presented in article.

6. PLOS authors have the option to publish the peer review history of their article (what does this mean?). If published, this will include your full peer review and any attached files.

Reviewer #1: **Yes: **ANDREW LANE

Reviewer #2: No

Reviewer #3: No

---

## [Author Response · Author response to Decision Letter 0]

1 May 2022

I have responded to all of the reviewers' comments. 

The emphasis in this revision was to specifically describe how control variables were handled. In connection with this attempt, I intend to further add information on the attributes of participants, research procedures, and analysis procedures to ensure the transparency of results. 

Thank you for your continued support.

---

## [Decision Letter · Decision Letter 1]

10 Jun 2022

PONE-D-21-17660R1

Individual-differences affecting emotion regulation behaviors of injured athletes: A retrospective quantitative study

PLOS ONE

Dear Dr. Tatsumi,

Thank you for submitting your manuscript to PLOS ONE. After careful consideration, we feel that it has merit but does not fully meet PLOS ONE’s publication criteria as it currently stands. Therefore, we invite you to submit a revised version of the manuscript that addresses the points raised during the review process.

 Please submit your revised manuscript by Jul 24 2022 11:59PM. If you will need more time than this to complete your revisions, please reply to this message or contact the journal office at plosone@plos.org. Please include the following items when submitting your revised manuscript:A rebuttal letter that responds to each point raised by the academic editor and reviewer(s). You should upload this letter as a separate file labeled 'Response to Reviewers'.A marked-up copy of your manuscript that highlights changes made to the original version. You should upload this as a separate file labeled 'Revised Manuscript with Track Changes'.An unmarked version of your revised paper without tracked changes. You should upload this as a separate file labeled 'Manuscript'.

We look forward to receiving your revised manuscript.

Kind regards,

Lorena Verduci

Staff Editor

PLOS ONE

Reviewers' comments:

Reviewer's Responses to Questions

**Comments to the Author**

1. If the authors have adequately addressed your comments raised in a previous round of review and you feel that this manuscript is now acceptable for publication, you may indicate that here to bypass the “Comments to the Author” section, enter your conflict of interest statement in the “Confidential to Editor” section, and submit your "Accept" recommendation.

Reviewer #1: (No Response)

Reviewer #2: (No Response)

Reviewer #3: All comments have been addressed

2. Is the manuscript technically sound, and do the data support the conclusions?

Reviewer #1: No

Reviewer #2: No

Reviewer #3: Yes

3. Has the statistical analysis been performed appropriately and rigorously? 

Reviewer #1: No

Reviewer #2: Yes

Reviewer #3: Yes

4. Have the authors made all data underlying the findings in their manuscript fully available?

Reviewer #1: Yes

Reviewer #2: No

Reviewer #3: Yes

5. Is the manuscript presented in an intelligible fashion and written in standard English?

Reviewer #1: Yes

Reviewer #2: Yes

Reviewer #3: Yes

6. Review Comments to the Author

Reviewer #1: I have read the responses to my comments which we intended to enhance the paper and give the author a window or him to see how the work comes across. In terms of the first response, that a macro perspective was taken. the work should be framed in a macro perspective and the theoretical nature of that approach explained. An especially serious point to make here is the influence of memory on the recall of emotions and how they are regulated, something made difficult when intense emotions are experienced on a regular basis. Many issues here in my view, largely dismissed by the author.

A second major issue is the interpretation of data. The argument that it is a causal analysis when it is a correlational analysis of memories over time but assessed at a single point in time. The effect of current mood on the recall of information could be large in some people and as such ambient mood might explain what information is recalled. Adding alpha coefficients misses the point.

In terms of what to add in terms of future reviews, I am somewhat stuck on what to say. I do not feel my last set of comments were taken seriously and so investing effort to offer comments on the new work seems unnecessary. However, I can recommend that I would think other readers might find similar issues as to the ones raised and as such question what the paper adds to the literature and from this perspective, feel that rejection is the only option I have available.

Reviewer #2: I thank the authors for the new version. However, I believe that some points made in the previous review were not addressed, for example indicating the study design and conclusions.

Reviewer #3: I have no further comments. All previous recommendations were fulfilled. In my opinion, the article meets the publication requirements of such a respected journal as PlosOne. Congratulations to the authors.

7. PLOS authors have the option to publish the peer review history of their article (what does this mean?). If published, this will include your full peer review and any attached files.

Reviewer #1: No

Reviewer #2: No

Reviewer #3: No

---

## [Author Response · Author response to Decision Letter 1]

4 Dec 2022

Response to academic editor

I apologize for the delay in submitting the revised manuscript.

In this revision, I have incorporated the comments of two reviewers (Reviewer 1 and 2). If the content is still insufficient, please point it out.

Thank you for your continued support.

Response to reviewer

Reviewer #1: I have read the responses to my comments which we intended to enhance the paper and give the author a window or him to see how the work comes across. In terms of the first response, that a macro perspective was taken. the work should be framed in a macro perspective and the theoretical nature of that approach explained. An especially serious point to make here is the influence of memory on the recall of emotions and how they are regulated, something made difficult when intense emotions are experienced on a regular basis. Many issues here in my view, largely dismissed by the author.

A second major issue is the interpretation of data. The argument that it is a causal analysis when it is a correlational analysis of memories over time but assessed at a single point in time. The effect of current mood on the recall of information could be large in some people and as such ambient mood might explain what information is recalled. Adding alpha coefficients misses the point.

<Responses>

I take the reviewer's comments as strict and correct comments. I think that problems such as memory problems that recall bias can occur and contradictions in causally analyzing data obtained at the same time are inherent in the design of this study. It is difficult to completely eliminate this problem, and this research addresses these problems in the following ways.

First, as a countermeasure against recall bias, in the questionnaire survey, as specified in the method, active student athletes were targeted and asked to pick up one injury experience (rehabilitation process) after entering university. Next, at the beginning of the survey, they were asked to give a general review (recollection) of their post-injury rehabilitation process, and were guided to respond to each scale. Although it is difficult to explain the validity of this method using existing theories, a procedure similar to the wash-out period that is conducted in clinical trials was taught at the start of the questionnaire survey. This is a common research procedure in this study design.

In addition, in order to address this bias, athletes who are currently injured, those who were unable to comeback to sports due to injury, or athletes who have returned to competition for less than a week have been excluded from the survey. In addition, as indicated in the procedure for consent to surveys, survey cooperation is not compulsory. Athletes who had difficulty recalling their injuries for various reasons were allowed to stop responding midway. As a result, there were no applicable athletes.

In other words, it can be assumed that the data analyzed in this study were obtained from participants who were able to face and recall their own injury experiences. Errors can occur in any study design. I believe that using large-scale data offsets such problematic effects. Although not perfect.

Next, regarding the direction of causality between variables, no matter how I devised the text of the instructions and questions, I think that the data obtained at the same time will not change. However, individual variables have a trait-level nature. Therefore, I interpret that it is possible to make causal inferences about how the three types of individual variables influence the choice of emotion regulation behavior.

If the above ideas and methods are naive, I think this is the limit of this study. I understand that the design of this study is not rigorous. However, due to the nature of the data dealing with the mind of injured athletes, a quantitative study based on retrospective data was first performed. There are no quantitative studies in this field. The conclusion of this study must not go beyond the realm of speculation, but at least I believe that I have provided a speculation that can be passed on to the next stage of study.

I won't take your comments lightly. Thank you very much.

Reviewer #2: I thank the authors for the new version. However, I believe that some points made in the previous review were not addressed, for example indicating the study design and conclusions.

<Responses>

I intended to address the above two points in the previous revision. However, I accepted that there was a lack of content, and rewrote everything.

First, the study design was specifically described at the beginning of the method. I also rewrote the conclusion. Please let me know if the content is still insufficient.

Your comment saved me. Thank you very much.

Reviewer #3: I have no further comments. All previous recommendations were fulfilled. In my opinion, the article meets the publication requirements of such a respected journal as PlosOne. Congratulations to the authors.

<Responses>

Your comment saved me. Thank you very much.

---

## [Decision Letter · Decision Letter 2]

2 May 2023

PONE-D-21-17660R2Individual-differences affecting emotion regulation behaviors of injured athletes: A retrospective quantitative studyPLOS ONE

Dear Dr. Tatsumi,

Thank you for submitting your manuscript to PLOS ONE. After careful consideration, we feel that it has merit but does not fully meet PLOS ONE’s publication criteria as it currently stands. Therefore, we invite you to submit a revised version of the manuscript that addresses the points raised during the review process.

We look forward to receiving your revised manuscript.

Kind regards,

Santiago Gascón, PhD

Academic Editor

PLOS ONE

Additional Editor Comments:

I consider that the authors comply with some of the recommendations raised by the reviewers.

However, they did not address central aspects of the review, such as the type of design and the conclusions that are drawn. Therefore, the study continues to present important weaknesses.

Indeed, the research addresses a subject as relevant as the emotional responses that can be felt after an injury. But the biggest limitation is that they do not raise this issue under a solid conceptual framework, in addition to the fact that, as the reviewers suggest, causal relationships cannot be established, but rather correlational ones.

This aspect should be underlined when the study is published.

If these aspects are considered again and are included as limitations, the study could contribute by showing its conclusions as a "theoretical trend" or "approach to a hypothesis" that, perhaps, could guide future research.

In addition to warning of the need to control aspects such as biases in the memory of emotions, affected by the state of mind when the participants are responding.

Reviewers' comments:

Reviewer's Responses to Questions

**Comments to the Author**

1. If the authors have adequately addressed your comments raised in a previous round of review and you feel that this manuscript is now acceptable for publication, you may indicate that here to bypass the “Comments to the Author” section, enter your conflict of interest statement in the “Confidential to Editor” section, and submit your "Accept" recommendation.

Reviewer #1: (No Response)

2. Is the manuscript technically sound, and do the data support the conclusions?

Reviewer #1: No

3. Has the statistical analysis been performed appropriately and rigorously? 

Reviewer #1: No

4. Have the authors made all data underlying the findings in their manuscript fully available?

Reviewer #1: No

5. Is the manuscript presented in an intelligible fashion and written in standard English?

Reviewer #1: Yes

6. Review Comments to the Author

Reviewer #1: The issues raised throughout the review process relate to the potential influence of current mood and recall of affective experiences and the use of causal language when the design of the research is correlational. The main response is that there are few studies. Whilst this is agreed, the study does not control for current mood as suggested by the authors in the revision. They needed to explain how this occurs but did not.

The second is the use of causal terminology - this remains. Sadly, and so the findings are overemphasised.

7. PLOS authors have the option to publish the peer review history of their article (what does this mean?). If published, this will include your full peer review and any attached files.

Reviewer #1: No

---

## [Author Response · Author response to Decision Letter 2]

12 Jun 2023

I would like to thank the editor and reviewers for their specific advice.

Thank you for your constructive feedback.

Thank you for your continued support.

---

## [Decision Letter · Decision Letter 3]

6 Nov 2023

Individual-differences affecting emotion regulation behaviors of injured athletes: A retrospective quantitative study

PONE-D-21-17660R3

Dear Dr. Tatsumi,

We’re pleased to inform you that your manuscript has been judged scientifically suitable for publication and will be formally accepted for publication once it meets all outstanding technical requirements.

Kind regards,

Santiago Gascón, PhD

Academic Editor

PLOS ONE

Additional Editor Comments (optional):

In my opinion, the authors have responded to the comments raised by the reviewers and I consider that the study should be published in PLOS One. Congratulations.

Reviewers' comments:

Reviewer's Responses to Questions

**Comments to the Author**

1. If the authors have adequately addressed your comments raised in a previous round of review and you feel that this manuscript is now acceptable for publication, you may indicate that here to bypass the “Comments to the Author” section, enter your conflict of interest statement in the “Confidential to Editor” section, and submit your "Accept" recommendation.

Reviewer #4: (No Response)

Reviewer #5: All comments have been addressed

Reviewer #6: (No Response)

2. Is the manuscript technically sound, and do the data support the conclusions?

Reviewer #4: Yes

Reviewer #5: Yes

Reviewer #6: Partly

3. Has the statistical analysis been performed appropriately and rigorously? 

Reviewer #4: Yes

Reviewer #5: Yes

Reviewer #6: Yes

4. Have the authors made all data underlying the findings in their manuscript fully available?

Reviewer #4: Yes

Reviewer #5: Yes

Reviewer #6: Yes

5. Is the manuscript presented in an intelligible fashion and written in standard English?

Reviewer #4: Yes

Reviewer #5: Yes

Reviewer #6: Yes

6. Review Comments to the Author

Reviewer #4: This is a well written paper. I have few minor comments.

Abstract

Typo, line 18, individual-differences.

Line 26-31. The conclusion is rather speculative in nature. I suggest rephrasing.

Introduction

Paragrahph 1 is lack of direction. Adding a synthesis will add clarity.

I appreciate the attention to detail by the author, however, line 77-99 could be consolidated into one paragraph.

Methods

Well construct.

N=205 is a sample or “population”. Needs to clarity information. If sample, would power calculation add justification for the number selected?

Results and Discussion

Well-presented and discussed.

Reference

References are rather “old” the latest being circa 2019. There are many current and related research papers that could be added. Please revise.

I cannot find ref #5. Fujii H. Approach to mind of injured athletes. Sportsmedicine. 2000; 12(2): 59-63. doi: 10.11501/4425909

Reviewer #5: This is an interesting article and I have no further comments to make. In my opinion, all recomendations were answered and because of this, the article have de publications requirements to be accepted.

Reviewer #6: The inappropriate use of causal language pointed out by Reviewer #1 remains, for example in lines 351-354 and further passages in the results and abstract sections. This issue should be addressed carefully to avoid misleading readers who might be skimming the study.

7. PLOS authors have the option to publish the peer review history of their article (what does this mean?). If published, this will include your full peer review and any attached files.

Reviewer #4: No

Reviewer #5: **Yes: **Rui Canário Lemos

Reviewer #6: **Yes: **Jonas Bischofberger

---

## [Editor Report · Acceptance letter]

13 Nov 2023

PONE-D-21-17660R3 

Individual-differences affecting emotion regulation behaviors of injured athletes: A retrospective quantitative study 

Dear Dr. Tatsumi:

I'm pleased to inform you that your manuscript has been deemed suitable for publication in PLOS ONE. Congratulations! Your manuscript is now with our production department. 

Kind regards, 

on behalf of

Dr. Santiago Gascón 

Academic Editor

PLOS ONE